# Online Gain Tuning Using Neural Networks: A Comparative Study

**Ashley Hill [1,*], Jean Laneurit [2], Roland Lenain [2] and Eric Lucet [1]**

1   List, CEA, Université Paris-Saclay, F-91120 Palaiseau, France
2   Centre de Clermont-Ferrand, Université Clermont Auvergne, Inrae, UR TSCF, F-63178 Aubière, France
*   Correspondence: ashley.hill@cea.fr

**Abstract:** This paper addresses the problem of adapting a control system to unseen conditions, specifically to the problem of trajectory tracking in off-road conditions. Three different approaches are considered and compared for this comparative study: The first approach is a classical reinforcement learning method to define the steering control of the system. The second strategy uses an end-to-end reinforcement learning method, allowing for the training of a policy for the steering of the robot. The third strategy uses a hybrid gain tuning method, allowing for the adaptation of the settling distance with respect to the robot's capabilities according to the perception, in order to optimize the robot's behavior with respect to an objective function. The three methods are described and compared to the results obtained using constant parameters in order to identify their respective strengths and weaknesses. They have been implemented and tested in real conditions on an off-road mobile robot with variable terrain and trajectories. The hybrid method allowing for an overall reduction of 53.2% when compared with a predictive control law. A thorough analysis of the methods are then performed, and further insights are obtained in the context of gain tuning for steering controllers in dynamic environments. The performance and transferability of these methods are demonstrated, as well as their robustness to changes in the terrain properties. As a result, tracking errors are reduced while preserving the stability and the explainability of the control architecture.

**Keywords:** mobile robotic control; path following; reinforcement learning; optimal control

## 1. Introduction

The field of agricultural robotics has enabled many advances in autonomous robotics. Indeed, an agricultural setting tends to have a wide variety of wheeled systems, soil types, and unique tasks. Finding methods to solve these problems allows for generalized solutions in robotics and automation in general. Improvements in sensors and sensor fusion methods allow for better perception [1], and improvements in sensor filtering for accurate localization [2] allow for better understanding of the robot's environment. Further advances in vision-based obstacle detection and navigation [3] can leverage perception to navigate and avoid obstacles correctly. Furthermore, by using machine learning-based computer vision, accurate crop segmentation can be achieved [4], allowing the desired tasks to be accomplished.

These methods are capable of identifying, locating, and navigating a mobile robot in an agricultural context autonomously. However, their adaptability to unexpected or unanticipated conditions still needs to be improved, as these aspects need to be explicitly defined in advance in the robot behavior algorithm. One possible improvement is to exploit the latest advances in machine learning, in particular deep reinforcement learning, to develop a method capable of deriving the desired behavioral algorithm for navigation by learning in a simulated environment for an extended period of time. Methods such as the one used for the Alpha-Go [5] experiment, have shown in practice their adaptability to novel situations, thanks to self-learning generalizations from the training environment, resulting in a more complete and adaptable behavior algorithm.

Classically, in order to control the steering of a wheeled robot in a path-following context, a control law should be employed or determined, such as the ones described in [6–9]. However, with advances in machine learning and deep reinforcement learning, new methods for determining control laws for a given task have been developed. These approaches require the complete replacement of the previous classical deterministic controller, which can be useful when no control law exists, as shown in the article [10]. Unfortunately, when a control law exists, there is no clear way to integrate both reinforcement learning and classical control.

Many studies have been conducted with deep reinforcement learning in agriculture [11–13], but some do not compare with existing classic control, and they do not consider the possible approaches that hybridize both classic predictive control and deep reinforcement learning.

As a result, this work is based on previous works by the authors [14], in which a gain tuning method based on deep reinforcement learning is proposed. In this paper, this method is re-described, and then compared with a classic predictive control law and a end-to-end deep reinforcement learning steering method. With a validation performed in a real world context in order to verify the simulated results.

Thus, a classical deep reinforcement learning steering approach for path following is presented, as well as a the hybrid control strategy by an online parameter tuning method of an existing control law using reinforcement learning. This work aims first to compare the performance of these two approaches in the context of off-road trajectory tracking, with an existing control law method is used as a baseline in a simulated context, followed by real world experiments in order to validate the simulated results. The goal of this is to compare the methods in order to distinguish what are the advantages and drawbacks of each of the three methods, and determine an optimal method for an off-road path tracking task.

In the next section, the details of the modeling method, the simulation, and the control law are defined in order train and simulate the methods described in the third section. Then using the methods, the simulated and real experiments are then derived in the forth section, of which the results are presented and analyzed in order to obtain comparative results. From these analysis and results, a discussion and the conclusions are then detailed in the fifth section, which shows the key aspects of each approach for the task off-road path tracking.

## 2. Mobile Robot Path Tracking Control

### 2.1. Assumption and Kinematic Description

In the framework of autonomous navigation, many approaches using dynamical model keep the bicycle representation, in order to reduce the number of parameters to be known [15], as the bicycle model is equivalent to a holonomic four wheeled robot with Ackerman steering. Since this work aims at considering the development of IA approaches to autonomously control a robot, a dynamical model is here exploited to observe and simulate the robot behavior. For this reason, we consider the simplest dynamical model (with as less parameters as possible) shown in [16], here applied to a single steering axle.

In this representation, the position of the center of gravity has to be known, as well as the robot mass $m$ and the moment of inertia $I_z$ along the vertical axis. The motion is supposed to be achieved on a flat ground, avoiding the consideration of a bank angle. Moreover, since we consider that the velocity is changing slowly, the longitudinal motion is not considered. As a result the contact forces acting at the tire ground patch are considered to be only oriented along a perpendicular axis with respect to tire's directions

Using these assumptions, the description of robot motion can be described as a dynamic model, with a constant cornering stiffness $C_F, C_R$, as depicted in [14].

When simulated and in real world experiments, a sliding angle observer [17] and a cornering stiffness observer [18] are used so to be able to observe these parameters for the approaches detailed in this paper.

## 2.2. Control Law Expression

Based on a kinematic model from the previously described modeling, a predictive controller name *Romea* for the front steering angle can be determined, as described in [6], which regulates the angular and lateral deviations. This pre-existing control expression from [6] allows the robot to reach the trajectory (i.e., ensuring the convergence of lateral deviation *y* to zero).

From this control expression, two parameters are introduced which allow for the adjustment of the robot's behavior:

- $K_p$: analogous to a proportional gain, it mainly defines the theoretical distance of convergence with respect to the lateral error.
- $K_d$: analogous to a derivative gain, it defines the theoretical distance of convergence with respect to the angular deviation.
- $H$: it defines the time in seconds for the lookahead horizon of the control law, in order to anticipate with respect to the action delay.

The predictive component of this controller is derived from the anticipation of the curvature servoing achieved by the controller, see [6] for a full description of the predictive aspects. From this predictive system, a third parameter is also defined, the prediction horizon denoted *H* of the future curvature. The horizon parameter and controller can in turn compensate for the convergence time of the steering with respect to the curvature (e.g., the action delay).

The control parameters used for this approach were determined experimentally over real world tests, in order to maximize the reactivity of the controller without oscillations. As show in the following Table 1.

**Table 1.** Control parameters used with *Romea*.

| Speed (m·s$^{-1}$) | 1.0 | 2.0 | 3.0 | 4.0 |
|---|---|---|---|---|
| $k_p$ (m$^{-2}$) | 1.0 | 0.7 | 0.4 | 0.4 |
| $k_d$ (m$^{-1}$) | 0.25 | 0.1225 | 0.01 | 0.01 |
| $H$ (s) | 0.5 | 0.5 | 0.5 | 0.5 |

Previously, the control parameters are tuned experimentally and manually, through an expert. These parameters rely on the desired velocity and expected grip conditions. In this paper, one of the proposed methods is used in order to adapt such parameters in an online fashion.

## 3. Steer Control with Reinforcement Learning

For the second approach, the control law is replaced with a neural network, which is trained by an optimizer. This is performed in a model free context, which means that the neural network only needs a target for optimization in order to train and converge. It is based on reinforcement learning, as the optimizer optimizes the neural network parameters, using a guided value that qualifies the desired behavior in a simulated environment, as opposed to learning a specific output from an input with supervised learning [19,20].

### 3.1. Neural Network Integration

For the NN controller method, the neural network directly controls the steering of the mobile robot's system, as shown in Figure 1 where the neural network takes the errors, curvature, and speed of reference, then returns the predicted steering control output.

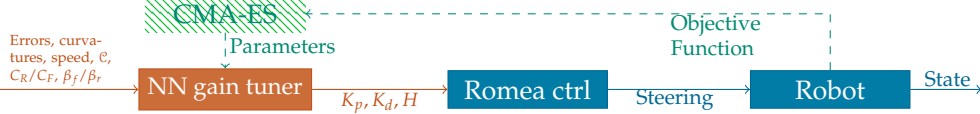

**Figure 1.** Overview of the RL steer method. Dashed section designates the optimizer that is only used for training.

The neural network is trained in a simulation using the Covariance Matrix Adaptation Evolutionary Strategy (CMA-ES) [21] optimizer depicted as the *Optimizer* in Figure 2, which takes as input the objective function value and returns the neural network parameters. The neural network takes as input the same information as an existing steering controller, which is the lateral error, angular error, curvature, and future curvature (20 sampled points over a $5s$ horizon of prediction, chosen in order to approximate the useful future curvature within 3 times the effective delay, while avoiding sub-sampling problem and without increasing the number of inputs by too much), speed, and the robot's steering state. In addition it also takes as inputs the dynamic parameters, such as the cornering stiffnesses ($C_R/C_F$), the sliding angles ($\beta_f/\beta_r$), and the sensors accuracy encoded in the Kalman covariance matrix ($\mathcal{C}$). From these inputs, the neural network is then expected to output the steering angle, with the speed control output being defined as constant (similarly to [22–26]). An Extended Kalman Filter (EKF) [27] is used as part of the *Observer* in order to filter the noise from the robot's sensors, determine the sensor accuracy, and improve the accuracy of the tracking. Following the EKF, a sliding angle observer is used [6] in order to estimate the front and rear sliding angles, which are needed to estimate the front and rear cornering stiffnesses. The latter are estimated then using a cornering stiffness observer [18]. The *Robot* block is the simulation, where the dynamic model of the robot described previously is used with a Runge–Kutta (RK4) integrator (when run in real world experiments, it is replace with the robot's interface), where the environment varies the grip conditions, maximum velocity, and trajectories for the robot. This was coded fully from scratch in custom C++ code with *libcmaes* for the CMA-ES implementation.

**Figure 2.** Overview of the gain tuning method. Dashed section designates the optimizer that is used only for training.

### 3.2. Training and Objective Function

This environment, however, induces problems with the Markovian hypothesis and the exploration. The expected time difference reinforcement learning methods such as Twin Delayed Deep Deterministic policy gradient (TD3) [28] or Proximal Policy Optimization (PPO) [29] do not converge, due to the inertia and action delay of the system. As such the CMA-ES optimizer was chosen, as optimizers similar to it were shown to have equivalent performance when compared to existing time difference reinforcement learning methods [30]. CMA-ES from [21] is an evolutionary strategy used for stochastic optimization of a problem space over a given objective function. In order to do this, CMA-ES calculates an estimate of the covariance matrix over each dimension of the problem space for the target objective function. When tested over the environment, this method shows the highest empirical performance out of the tested methods for training the neural network.

The neural network used is a fully connected neural network that is composed of three hidden layers of 64, 128, and 32 neurons respectively, (approx 15,000 parameters), with 23 inputs, and 1 to 3 outputs (depending on the required outputs). These values where determined in order to maximize the predictive capabilities of the neural network

(as shown in [31]), while minimizing the issues linked with high dimensional search spaces for CMA-ES.

Using the simulated model of the robot, the neural network (NN) and the optimizer CMA-ES, a trained neural network is derived, after 20,000 iterations of CMA-ES have been achieved, which represents over 2 years of simulated time and 24 h of wall time.

The objective function is defined as a composition of different targets. Indeed keeping a minimal distance to the trajectory is not sufficient as an optimization target, as it does not prevent from oscillations when the lateral error is low enough. As the function needs to return a scalar value from a set of sampled state vectors, an integration must take place. As such, a discrete integration over the curvilinear abscissa is performed, in order to avoid any side effects due to speed modulation. The result of the integration is then normalized over the length of the trajectory in order to keep the objective function values consistent between each trajectory. The objective function is defined as a compromise between two sub objectives: minimizing the lateral error, and minimizing the steering error. The results can be described as such:

$$obj_1 = \frac{1}{s_N} \sum_{i=0}^{N} \left[ |k_{yi} y_i| + k_{steer} |Lc(s) - \tan(\delta_{Fi})| \right] \Delta s \tag{1}$$

where the objective function for training is defined with an allowed error corridor of $y_{lim} = 0.20$ m. It was found experimentally though trial and error that a $k_{steer} = 3$, $k_{y\,low} = 1$, and $k_{y\,high} = 10$ returned a trained model with the ideal performance. Where $N$ is the number of samples recorded over the trajectory, $s_N$ is the length of the trajectory, and $k_{yi}$ is a dynamic objective function parameter, that will change the lateral error penalty if the lateral error exceeds a given limit $y_{lim}$:

$$k_{yi} = \begin{cases} k_{y\,low} & \text{if } |y_i| \leq y_{lim} \\ k_{y\,high} & \text{else} \end{cases}$$

where $ds$ denotes the rate of change of the curvilinear abscissa over time:

$$\Delta s = vs. \cos(\tilde{\theta}) \Delta t$$

This objective function is then used as a target for the optimization method, namely the evolutionary strategy called CMA-ES [21].

## 4. Online Control Parameter Tuning

Replacing an existing control law with a neural network can be considered excessive. As the expertise contained in the control law needs to be completely rediscover by the neural network empirically, which naturally can cause a drop in performance and stability when compared. As such, the proposed approach is a hybrid method called NN gain tuner, which consist of preserving the existing control law and using a neural network to determine the control parameters $K_p$, $K_d$, and $H$ in an online fashion. This should allow for the preservation of the controllers behavior, while adjusting it's reactivity and behavior to the environment, which should improve it's performance considerably.

Due to the similarities between the control task described previously and the proposed approach, the objective function, neural network architecture and optimizer are identical. Where only the output of the neural network is changed as show in the following.

### Neural Network Integration

For this approach, the neural network predicts the control parameters in real-time. In this case, they are the steering control gains and horizon, which are then passed to the controller before it calculates the steering angle. This is shown in Figure 2 where the neural network takes as inputs the errors, curvature, and speed, then returns the steering control

gains and horizon. The control law that is used in tandem is the Romea control law, in order to preserve the comparability with the previous methods.

The neural network is trained in a simulation using the CMA-ES optimizer, which takes as input the objective function value and returns the neural network parameters. The neural network takes as input the same information as an existing steering controller, which is the lateral error, angular error, curvature, future curvature, speed, and the robot's steering state. In addition it also takes as inputs the dynamic parameters, such as the cornering stiffnesses, the sliding angles, and the Kalman covariance matrix. From these inputs, the neural network is then expected to output the control parameters. As for the previous method, an Extended Kalman Filter (EKF), a sliding angle observer, and a cornering stiffness observer are used.

In order to compare the NN gain tuner method, an existing alternative model based deterministic gain tuning method is also employed called model gain tuner, that is detailed in [14]. It consist of using a dynamic model, in order to adjust the control gain with respect to the time to convergence of the angular speed.

## 5. Results

The metric used for the analysis of the results is the surface error:

$$A_{\text{error}} = \sum_{i=0}^{N} \left| v_i \cos(\tilde{\theta}_i) \left( y_i + \frac{v_i \sin(\tilde{\theta}_i)\Delta t}{2} \right) \right| \Delta t \qquad [\text{m}^2] \quad (2)$$

This is performed in order to validate and compare the performance of the different tested methods and parameters, without resorting to the objective function. Indeed, when a reinforcement learning agent trains to optimize a function, it is possible that the said agent might exploit the objective function in order to minimize it, without achieving the desired behavior. As such, using a different metric to measure performance allows for minimal bias when comparing the methods.

### 5.1. Simulated Results

A set of simulated runs was computed using the training trajectories. From this, Table 2 was obtained.

It describes the Surface error from Equation (2), for each method at all the speeds and trajectories used during the training, with an initial error of 1 m. The underlined and bold values mean that the result is significant and has a *p*-value below $10^{-3}$, determined using the Welch-t test [32].

Overall, the average surface error for NN gain tuner is significantly lower than the other methods at 6.88 m$^2$, whereas the surface error for *Romea* was 14.7 m$^2$ (a 53.2% reduction). The surface error for the model gain tuner method was 9.51 m$^2$ (a 27.7% reduction), and the surface error for the NN gain tuner method was 10.2 m$^2$ (a 32.5% reduction). From this table, more specific strengths and weaknesses can be observed. The NN gain tuner was able to match or exceed the performance of the NN controller and *Romea* methods in all cases. It was also able to match the performance of the model gain tuner method at lower speeds and exceed it at 3 m·s$^{-1}$ & 4 m·s$^{-1}$, thanks to the additional information that the NN gain tuner is able to leverage when compared to the model gain tuner at higher speeds. Specific edge cases can be observed however with the *line* and *estoril5* trajectories at lower speeds, as the model gain tuner is able to slightly exceed the performance of the NN gain tuner method, which are likely due to the accuracy of model used in the model gain tuner having better predictive capabilities at lower speeds.

In order to better interpret and understand the neural network, a gradient based Feature importance analysis can be used to determine which inputs where useful, and quantify the utility of each input with respect to each output. See [33] for details on the theory and implementation of the gradient based feature importance analysis for the neural network. Using the feature importance analysis, the following results are obtained.

**Table 2.** Surface error in (m²)of each method at all the speeds and trajectories used during training, with an initial error of 1 m. Bold and underlined numbers show that the results are significant, green shows the best result, and red show the worst result.

|  |  | estoril5 | estoril7 | estoril910 | spline5 | line |
|---|---|---|---|---|---|---|
| 1 m·s⁻¹ | Romea | **4.70** (±0.33) | **5.33** (±0.63) | **33.12** (±0.91) | **7.15** (±1.01) | **4.35** (±0.08) |
|  | model gain tuner | **2.92** (±0.26) | 3.11 (±0.53) | **27.76** (±1.20) | **4.38** (±1.17) | **2.89** (±0.12) |
|  | NN controller | **4.77** (±0.35) | **5.14** (±0.63) | **24.45** (±1.15) | **6.67** (±1.57) | **4.74** (±0.30) |
|  | NN gain tuner | 2.98 (±0.22) | 3.09 (±0.51) | 22.86 (±1.54) | 3.80 (±0.86) | 2.97 (±0.09) |
| 2 m·s⁻¹ | Romea | **6.50** (±0.61) | **8.06** (±0.99) | **39.17** (±1.79) | **10.82** (±1.16) | **5.52** (±0.10) |
|  | model gain tuner | **3.53** (±0.72) | **4.46** (±1.30) | **33.44** (±2.83) | **6.68** (±1.69) | **2.99** (±0.22) |
|  | NN controller | **5.16** (±0.44) | **5.87** (±0.96) | **20.80** (±1.37) | **7.78** (±1.38) | **4.36** (±0.21) |
|  | NN gain tuner | 3.38 (±0.41) | 3.88 (±0.94) | 19.42 (±2.08) | 5.18 (±1.34) | 3.11 (±0.14) |
| 3 m·s⁻¹ | Romea | **12.31** (±1.22) | **16.09** (±1.58) | **64.85** (±5.14) | **19.94** (±1.95) | **9.21** (±0.15) |
|  | model gain tuner | **4.82** (±1.83) | **6.78** (±3.05) | **46.26** (±9.07) | **9.94** (±3.36) | 3.32 (±0.44) |
|  | NN controller | **6.22** (±0.57) | **7.19** (±0.88) | **23.30** (±2.03) | **9.56** (±1.59) | **5.12** (±0.30) |
|  | NN gain tuner | 3.97 (±0.81) | 5.01 (±1.54) | 20.81 (±3.44) | 7.99 (±2.32) | 3.27 (±0.23) |
| 4 m·s⁻¹ | Romea | **13.44** (±1.59) | **18.13** (±2.37) | **68.75** (±9.39) | **22.68** (±3.91) | **9.13** (±0.17) |
|  | model gain tuner | **6.96** (±3.94) | **10.02** (±5.53) | **73.06** (±24.10) | **16.13** (±6.57) | 3.66 (±0.72) |
|  | NN controller | **9.12** (±1.31) | **10.51** (±2.00) | **35.97** (±6.98) | 13.99 (±5.23) | **7.83** (±1.01) |
|  | NN gain tuner | 5.41 (±2.41) | 6.93 (±2.52) | 30.09 (±12.95) | 12.02 (±18.20) | 3.59 (±0.30) |

From Figure 3, the inputs that contribute the most the outputs of the NN gain tuner method are in order of importance the rate of change of the angular error denoted $d\tilde{\theta}/dt$. The rate of change of the lateral error denoted $dy/dt$, which contribute a total of 40% of the variations of the outputs of the NN gain tuner method. The rest of the inputs seems to have a uniform importance distribution, which implies that most of the inputs are useful for predicting the outputs of the neural network, which in turn demonstrates when compared to the previous feature importance analysis that the addition of these inputs allows for a richer prediction of the control gains and horizon.

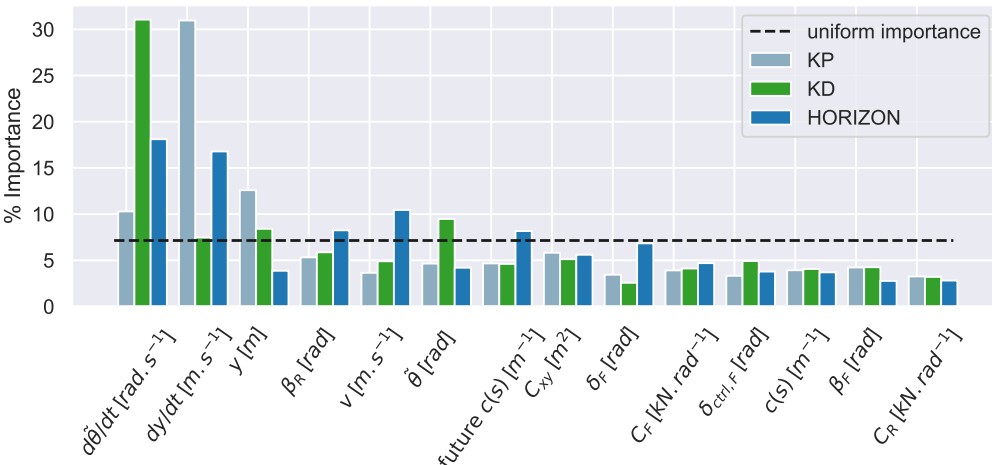

**Figure 3.** The feature importance for the NN gain tuner method for each input, denoted in % of importance.

This shows that the NN gain tuner is correcting the control gains output using most of the inputs available to it. This allows for a more complex prediction of the gain and horizon, which allows a higher level of performance with respect to the objective function and metrics as shown previously. Furthermore, the neural network is able to take into account the sensor accuracy denoted $\mathcal{C}_{xy}$ as shown previously, as it is considered more important as the cornering stiffnesses denoted $C_F$ & $C_R$ that is used by the model gain tuner shown previously. These factors allows the NN gain tuner to outperform the model gain tuner when the grip conditions and the sensor accuracy change over time.

### 5.2. Real World Results

Due to the poor performance and the potentially high instability of the NN controller method to novel environment, it was not exploitable in real world conditions when tested. Furthermore, when tested the robot exhibited dangerous behavior, which prevented its experimental runs due to safety constraints not being able to be met, such as a oscillatory positive feedback over the lateral error which induced high lateral errors and unpredictable motion.

The following experimental results are based on the same experimental tests performed for the authors previous paper [14], with a new analysis to validate the behavior implied in the simulated results in a real-world setting.

### 5.2.1. The RobuFast Robotic Platform

The proposed methods have been tested on the *RobuFAST* robotic platform (Figure 4). The robot's mass is about 420 kg, has a vertical moment of inertia of 300 kg·m², a wheelbase of 1.2 m from the center of each axles, a center of mass at 0.625 m from the center of the rear axle, and a front steering response time of 0.45 s. The platform runs on the Robotic Operating System (ROS) middleware with a control frequency of 10 Hz. It has an Inertial Measurement Unit (IMU) and a Real Time Kinematics Global Positioning System (RTK-GPS) which updates the observers and state estimators every 10 Hz. The sliding angles observer is tuned for a settling time of 0.5 s, and the cornering stiffness observer is tuned for a settling time of 1.5 s.

The RobuFAST robot, is as robot issued from the FAST project. It has been designed as an experimental platform, that has been modified in order to reach $8.0 \text{ m·s}^{-1}$. The following Table 3 denotes the characteristics of the RobuFAST platform:

**Table 3.** The given RobuFAST characteristics table, for reference.

| | |
|---|---|
| Mass ($m$) | 420 Kg |
| Vertical moment of inertia ($Iz$) | 300 Kg·m² |
| Wheelbase ($L$) | 1.2 m |
| Front Wheelbase ($LF$) | 0.575 m |
| Max steering | 20° |
| Steering action delay (Pure) | 0.2 s |
| Steering action delay (Rise time) | 0.25 s |
| Steering action delay (Total) | 0.45 s |
| Max speed | $8.0 \text{ m·s}^{-1}$ |
| Max acceleration | $1.5 \text{ m·s}^{-2}$ |

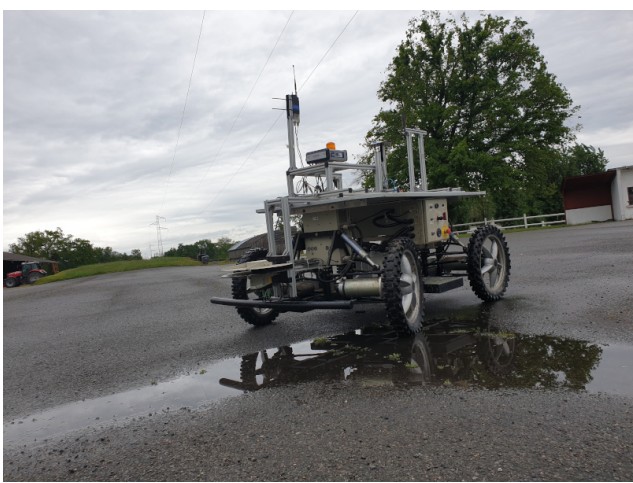

**Figure 4.** The RobuFAST robotic platform.

The experiments on the platform were run over a sunny warm day and with cloudless weather. With the following trajectories shown in Figure 5.

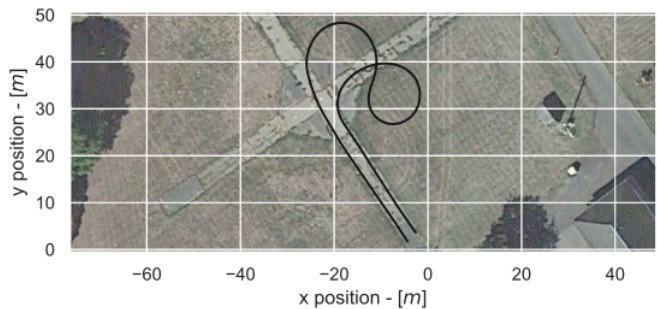

**Figure 5.** The trajectory.

The robot starts with an initial lateral error, with a straight line to observe the stabilization from said error. A sequence of sharp corners to stress the control system. Then, a straight line to observe the stabilization from the corners. The trajectory starts on concrete, reaches grass on the corners, and the last straight line on concretes. As such, it has relatively good grip, and transition on the type of ground ($C_r$ & $C_f \sim 15{,}000$ N·rad$^{-1}$).

### 5.2.2. The Results

The following results were obtained at 3 m·s$^{-1}$ over the first trajectory.

As shwon in Figure 6, the error over the entire trajectory is the lowest with the NN gain tuner method, with the model gain tuner method obtaining some significant errors, and where the expert gain had the largest error. However, NN gain tuner does oscillate around 0 m of lateral error.

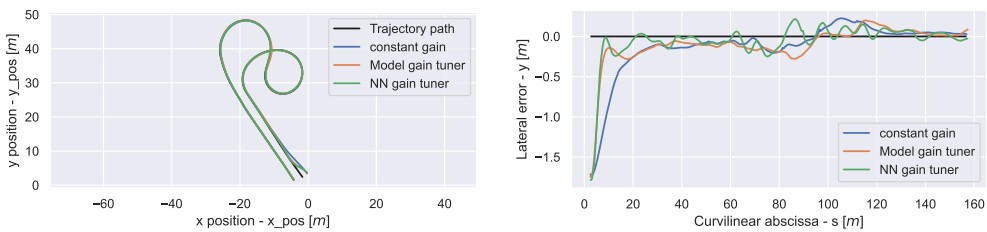

**Figure 6.** The trajectory (**left**) and the lateral error (**right**), over the total trajectory.

A result that is reflected clearly in the surface error in Figure 7. Where the NN gain tuner method reaches 17.7 m² or a 29.2% reduction in the surface error, and where the model gain tuner method reaches 19.4 m² or a 22.5% reduction in the surface error, when compared with the Romea controller which reaches 25.0 m² for the surface error. If we do not include the starting error, the results are 12.8 m² (0.696% increase) and 13.0 m² (2.76% increase), respectively, when compared with the Romea controller which reaches 12.7 m² for the surface error.

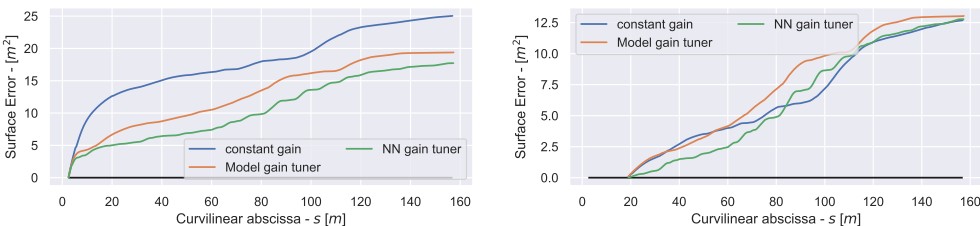

**Figure 7.** The surface error $A_{\text{error}}$ (**left**) and the surface error $A_{\text{error}}$ after the initial lateral error (**right**).

The neural network gains shown in Figure 8 seem to be much higher than the expert and model gain tuner gains. However a strong modulation between $k_p$ and $k_d$ can be observed, allowing for the method to dynamically update the damping factor $\xi$, which in turn explains the higher performance using higher gains, because a damping factor below $\xi < \frac{\sqrt{2}}{2}$ is considered unstable but can allow for a faster convergence to the trajectory if used correctly.

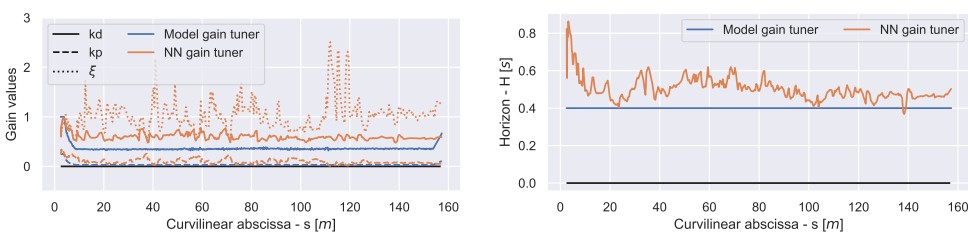

**Figure 8.** The gains (**left**) and the horizon (**right**).

Overall, we can see that the neural network parameter tuning method, is capable of matching and even outperforming the proposed model gain tuner method at 3 m·s⁻¹ when tested in real world conditions, which confirms the performance seen in simulation. However, it should be noted that the training of the neural network was unable to exceed 4 m·s⁻¹. It is assumed that this limitation is due to the task being too difficult over the training trajectories at higher speed. As such, the future experiments also modulate the maximum speed.

## 6. Conclusions

In this paper, three methods have been described and their performances analyzed. The first one is a classical control law existing in the state of the art, tuned with constant gains. The second is an approach typically explored in machine learning, which consists of replacing the control law by a trained neural network. Furthermore, the third approach is a hybrid of the previous methods, where an existing control law is tuned in real time using a neural network. The simulation results showed that the hybrid control approach outperforms the other two approaches by 53.2% and outperforms the existing gain-tuning approaches by 27.7%.

Overall, the real-world experimental results show that the performance observed in simulation is transferable to the real world. This is demonstrated by the performance of the NN gain tuner which significantly outperforms the constant gain and model gain tuner.

This is partly due to the strong modulation of the damping factor $\xi$, where the NN gain tuner punctually selects a $\xi < \frac{\sqrt{2}}{2}$ considered as oscillatory, in order to converge more quickly to the desired setpoint.

Nevertheless, as can be seen by modulating the gains, the performances of the methods depend on a defined gain/speed couple. Indeed, the optimal gain varies according to the speed, and conversely the optimal speed depends on the value of the control gains. Moreover, there are situations where there are no valid gains for certain speeds (for example, a speed that is too high for a turn that causes a spin-out). Thus, future work will consider the simultaneous tuning of speed and gains to further improve performance and allow for greater adaptability to the environment.

**Author Contributions:** Conceptualization, A.H., E.L. and R.L.; methodology, A.H., E.L. and R.L.; software, A.H., J.L. and R.L.; validation, A.H., E.L. and R.L.; formal analysis, A.H.; investigation, A.H., J.L., E.L. and R.L.; resources, A.H., J.L. and R.L.; data curation, A.H., E.L. and R.L.; writing—original draft preparation, A.H. and E.L.; writing—review and editing, E.L. and R.L.; visualization, A.H.; supervision, E.L. and R.L.; project administration, A.H., E.L. and R.L.; funding acquisition, N/A; All authors have read and agreed to the published version of the manuscript.

**Funding:** This research received no external funding.

**Data Availability Statement:** The data presented in this study are available on request from the corresponding author. The data are not publicly available due to organization constraints.

**Acknowledgments:** This publication was made possible by the use of Factory-IA cluster and was financially supported by the Ile-de-France Regional Council. It received the support of the French government research program "Investissements d'Avenir" through the IDEX-ISITE initiative 16-IDEX-0001 (CAP 20-25) and IMobS3 Laboratory of Excellence (ANR-10-LABX-16-01).

**Conflicts of Interest:** The authors declare no conflict of interest.

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
