# Peer review of "Online Gain Tuning Using Neural Networks: A Comparative Study"

_agriengineering, doi:10.3390/agriengineering4040075_

Round 1

Reviewer 1 Report

In abstract, the major quantitative results (increase and/or errors of tracking performance) could be added.

It is argued in Introduction that reinforcement learning is "a latest advance in machine learning" - on the contrary, this method has been known for few decades, please specify the presentation and usage of RL in the text.

Introduction now minimally reviews the current control strategies and other methods applied to the tracking task of vehicles. This should be further addressed by the authors.

There are many acronyms in the text that should be written open for the first time at least.

After equation 4, the point seems to be missing. The units are mostly connected to related numbers, there should be a space between them, but not with %-mark.

Abstract and Conclusions together with main text should be consistent concerning the content of the manuscript. Now for example, in Abstract there is mention on two proposed control methods, whereas in Conclusion there are three methods. Please, clarify and unify the notation and number of the methods presented.

The results of real-world experiments are mostly presented in Figures, qualitatively. To achieve more significant conclusions, a numeric comparison could be further added with errors as with training stage, besides the percentages. 

What kind of software and hardware were utilized for the controllers? These could be reported too.

Author Response

Thank you for your time reviewing this paper and for the comments that will help improve it's understanding and quality.

  • In abstract, the major quantitative results (increase and/or errors of tracking performance) could be added.

Indeed this is an excellent point. An indicator of the overall reduction of the tracking errors has been added to the abstract, with the following sentence: "The proposed method allowing for an overall reduction of 53.2% when compared with a predictive control law."

  • It is argued in Introduction that reinforcement learning is "a latest advance in machine learning" - on the contrary, this method has been known for few decades, please specify the presentation and usage of RL in the text.

This was a slight oversight for the authors, deep reinforcement learning such as A2C and PPO where implied. The sentence was corrected to: "... with advances in machine learning and deep reinforcement learning ..."

  • Introduction now minimally reviews the current control strategies and other methods applied to the tracking task of vehicles. This should be further addressed by the authors.

Additional references to control strategies have been added to the introduction, in order to improve the well-roundedness of the literature study.

  • There are many acronyms in the text that should be written open for the first time at least.

Many of the acronyms have been fully written, such as CMA-ES, PPO, TD3 in section 3.1,  ROS, IMU, RTK-GPU in section 5.2. The FAST acronym unfortunately is a noun and does not stand for any fully written sentence

  • After equation 4, the point seems to be missing. The units are mostly connected to related numbers, there should be a space between them, but not with %-mark.

Unfortunately I do not seem to find the specified issue, as below eq.4, there are no "%" symbols present. Equation 4 describes an objective value function, and as such it tends to be dimensionless, as it is a minimization of multiple criteria that are not always physically linked.

  • Abstract and Conclusions together with main text should be consistent concerning the content of the manuscript. Now for example, in Abstract there is mention on two proposed control methods, whereas in Conclusion there are three methods. Please, clarify and unify the notation and number of the methods presented.

The paper has been corrected in order to address this mistake. Now the introduction and the abstract both describe the three methods addressed in the paper.

  • The results of real-world experiments are mostly presented in Figures, qualitatively. To achieve more significant conclusions, a numeric comparison could be further added with errors as with training stage, besides the percentages. 

Indeed, this is critical information which would allow for addition comparatives for the readers. As such, the values of the surface error have been given along side their respective percentage values.

  • What kind of software and hardware were utilized for the controllers? These could be reported too.

The software used was mostly custom C++ code with libcmaes used as the CMA-ES implementation. This was clarified in the end of section 3.1.

Reviewer 2 Report

ŸYou need to explain reinforcement learning in more detail.

ŸYou should explain the related work about agricultural robotics in more detail.

ŸWhy did you choose only reinforcement learning methods? it seems better to explain and compare classical methods additionally(not deep learning).

ŸWhy did you use the bicycle representation by the dynamical model? (Actual testing didnt use bicycle as shown Fig. 6.)

Ÿ Please describe the simulation environments.

Ÿ It seems necessary to explain in more detail why the test in real world conditions was impossible for NN Controller.

Ÿ The capital letters of the title are not unified.

Ÿ Overall, the size of the figures is small, so that it is difficult to understand the content.

Ÿ Overall, it is necessary to change the location of the tables or figures in a consistently.

Author Response

Thank you for your time reviewing this paper and for the comments that will help improve it's understanding and quality.

  • You need to explain reinforcement learning in more detail.

Section 3 has been changed in order to clarify how the approach is based on reinforcement learning along with additional citation in case the readers would like to dive deeper in the theory of reinforcement learning:

"It is based on reinforcement learning, as the optimizer optimizes the neural network parameters, using a guided value that qualifies the desired behavior in a simulated environment, as opposed to learning a specific output from an input with supervised learning."

  • You should explain the related work about agricultural robotics in more detail.

An additional paragraph has been added to the introduction in order to address the papers that use deep reinforcement learning in agriculture.

  • Why did you choose only reinforcement learning methods? it seems better to explain and compare classical methods additionally(not deep learning).

Due to a introduction and abstract that were not clear, many reviewers were confused with the content of the paper. This has been addressed to explain clearly that three methods are compared: Deep reinforcement learning, Classic model based predictive control, and a hybrid of Deep RL and classic control.

Furthermore, previous works by the authors have explored gain tuning as described with the hybrid approach, but without using any deep learning with good results, but below what was achieved with the RL hybrid method. Due to size constraints this method was not explored in the paper, but would have been added if it was possible as it is an interesting comparative.

  • Why did you use the bicycle representation by the dynamical model? (Actual testing didn’t use bicycle as shown Fig. 6.)

This is an excellent question that was not explained in the paper. When an Ackerman steering is applied to a holonomic four wheeled robot, its model is reducable to a bicycle model, allowing for simpler modeling equations. This has been added to section 2.1.

  • Ÿ Please describe the simulation environments.

Indeed a description of the simulation was missing, it has been added to the section 3.1 which describes the blocks for the training of the methods. Notably:

"The Robot block is the simulation, where the dynamic model of the robot described previously is used with a Runge-Kutta (RK4) integrator (when run in real world experiments, it is replace with the robot's interface), where the environment varies the grip conditions, maximum velocity, and trajectories for the robot."

  • Ÿ It seems necessary to explain in more detail why the test in real world conditions was impossible for NN Controller.

The key reason that was omitted due to lapse on the author's part, is for safety reasons. Indeed the robot was very dangerous and unpredictable when running with the NN Controller, due to the differences between the trained simulated environment and the real world environment. This has been clarified now in the paper section 5.2:

"Furthermore, when tested the robot exhibited dangerous behavior, which prevented its experimental runs due to safety constraints not being able to be met, such as a oscillatory positive feedback over the lateral error which induced high lateral errors and unpredictable motion."

  • Ÿ The capital letters of the title are not unified.

This is a clear oversight of the authors, all the titles are now correctly capitalized

  •  Overall, the size of the figures is small, so that it is difficult to understand the content.

The figures have been ajusted in order to be as large as possible within the bounds of the journal's format

  • Ÿ Overall, it is necessary to change the location of the tables or figures in a consistently.

The indentation of the table 2, table 3, and figure 6 have been ajusted in order to be in line with the journal's formatting.

Reviewer 3 Report

It is very interesting paper written in excellent manner. However, please clarify what is the main difference between this paper and already published your paper (ref. 23)

Online Tuning of Control Parameters for Off-Road Mobile Robots: Novel Deterministic and Neural Network-Based Approaches?

Author Response

  • It is very interesting paper written in excellent manner. However, please clarify what is the main difference between this paper and already published your paper (ref. 23)

    Online Tuning of Control Parameters for Off-Road Mobile Robots: Novel Deterministic and Neural Network-Based Approaches?

Thank you for your time reviewing this paper and for the comments that will help improve it's understanding and quality.

Indeed both papers discuss the gain tuning of a controller for a mobile robot, however the previous paper describes the performance of a gain tuning method when dynamic parameters are available. Where as the proposed paper describes the comparative between a gain tuning method and existing control strategies for mobile robotics, as in comparing reinforcement learning and predictive control with the gain tuning method.   This paper shows that deep reinforcement learning alone is outperformed by a gain tuning approach using deep reinforcement learning.   An additional paragraph was added to the introduction, in order to better understand the differences from the previous paper, and indeed show that there are some correlations as it was taken as a base.

Reviewer 4 Report

The authors propose the use of neural networks for tuning the machine learning algorithms used to control the autonomy trajectory. The work presents a major technical contribution, and its work is support by a serie of experiments.

Some drackbaws in the work, is that the authors in their title propose a comparative study, but they only make a comparison between the algorithms proposed by them, they should be considered to include how their work differs from other publications in the same line. 

Is not clear for readers the research question to be resolved, and how their proposal contribute to resolve that question.  

Introduction section need to be improved.  More details of the research methodology and the scope of reseach need to be included.

In the last  paragraph of introduction section, the  information of the topics to be cover in the next sections  is very general.  Although the idea for that  paragraph is not be very extensive, it is required to be more clear about  what is covered in each section, and how each section connects with the previous ones, and how the cover in each section supports the resolution of the research problem (question). 

Author Response

Thank you for your time reviewing this paper and for the comments that will help improve it's understanding and quality.

  • Some drackbaws in the work, is that the authors in their title propose a comparative study, but they only make a comparison between the algorithms proposed by them, they should be considered to include how their work differs from other publications in the same line. 

The "comparative study" here was meant in the sense of comparing Reinforcement learning with classic control and with a hybrid of the two, in order to ascertain the performance for off-road mobile robots. However, "comparative study" does imply a broad study of the field, which is not the case. As such, the abstract has been amended, in order to clarify this. "Three different approaches are considered and compared for this comparative study:"

  • Is not clear for readers the research question to be resolved, and how their proposal contribute to resolve that question. 
  • Introduction section need to be improved.  More details of the research methodology and the scope of reseach need to be included.

These point are indeed important in order to contextualize the paper, and clearly show the goal. These have not been done due to an oversight, and have been added to the 5th paragraph of the paper, with an additional 4th paragraph in order to contextualize with previous works.

  • In the last  paragraph of introduction section, the  information of the topics to be cover in the next sections  is very general.  Although the idea for that  paragraph is not be very extensive, it is required to be more clear about  what is covered in each section, and how each section connects with the previous ones, and how the cover in each section supports the resolution of the research problem (question). 

Thank you for this explanation. The last paragraph as written seems to be a fusion of the next sections and a descriptive of what is done in the paper, which is indeed not ideal. It has been rewritten as such:

"In the next section, the details of the modeling method, the simulation, and the control law are defined in order train and simulate the methods described in the third section. Then using the methods, the simulated and real experiments are then derived in the forth section, of which the results are presented and analyzed in order to obtain comparative results. From these analysis and results, a discussion and the conclusions are then detailed in the fifth section, which shows the key aspects of each approach for the task off-road path tracking."

Round 2

Reviewer 3 Report

I am sorry, but most parts of the paper including figures were already published in your previous paper. The novel parts of the paper are minor and should be separated from the already published parts. In its current form the paper cannot be published.

Author Response

Thank you for your comments and clear review, we understand the reviewer's concern and, as such, we will detail the similarities and differences between the two articles.

The previous paper demonstrated a new method for online gain tuning using a model rather than a neural network as our previous work details.
In fact, this paper reuses the methods presented in the previous paper without detailing them again. Specifically, the neural network based gain tuning method and the model based gain tuning method.

However a novel method of not just gain tuning but a complete AI-based control scheme has been detailed, implemented, and tested with new simulated results. The key contribution of this paper is the new comparative results showing that the standard RL community approach is outperformed by the hybrid NN gain tuning method, where the RL community classically replaces the entire control law with a neural network.

We also recognize that the title of the document is poorly worded. We are considering:
    "Comparison of neural network-based control approaches in an agricultural context"

The introduction and abstract are unique to each paper, as they present different contexts. Indeed, the introduction to this article details AI-based control and lays the groundwork for the comparison between the full AI-based control scheme and the previous method.

The headings in Section 2 are the same because they detail the same control law and context. 

Next, the novel NN controller is detailed in Section 3.

Section 4 resume the gain tuning method presented in the previous paper.

Section 5.1 details the new simulated results (the previous paper did not contain simulated results). In addition to the new key analysis (using p-values) present in the simulated results comparing four control strategies on different speeds and path geometries, in order to obtain more relevant results. This allowed in practice to obtain a feature importance results on a more exhaustive data set and as representative of the whole dynamic range as possible by the robot.

Section 5.2 details the results on a real world robot and indeed uses the same dataset and therefore the same curves, however the interpretation of the results was done to confirm the analysis of the simulated results and not as an independent analysis (as was done in the previous paper). From this data, a new trajectory plot of the methods as seen from above and a new control horizon plot were also added in order to better understand the robot behavior.

Section 6 concludes with simulated and real-world results that shed additional light on the issue, in particular why the NN gain setting performs better than the model gain setting, why a NN controller is not ideal for the given use case, and why additional work on velocity fluctuations is needed.
In order not to mislead readers, the following paragraph has been added to the beginning of Section 5.2, which discusses real-world experimentation:
"The following experimental results are based on the same experimental tests performed for the authors previous paper [14], with a new analysis to validate the behavior implied in the simulated results in a real-world setting."

A change in section 5.2 was made to show that the NN controller was tested but had unsafe behavior.

We hope that these changes and details will be acceptable to the reviewers, and if not, please specify which specific parts of the paper that have been detailed are not acceptable for publication.

Round 3

Reviewer 3 Report

Dear authors,

Figure 1 is equivalent to Figure 1 in your previous paper [14].

Figure 2 is equivalent to Figure 2 in your previous paper [14].

Figure 3 appears as a part of Figure 5 in your previous paper [14].

Figure 4 mimics Figure 4 in your previous paper [14] with less details.

Figure 7 presents the same trajectories as Figure 7 in your previous paper [14].

Figure 8 (right) is equivalent to Figure 8(a) in your previous paper [14].

Figure 11 (right) is equivalent to Figure 9(a) in your previous paper [14].

So, among 11 figures, 7 figures present already published information.

Equation (1) is equivalent to Equation (2) in your previous paper [14]. The equation defines already defined model, so the reference is needed.

In my opinion, the paper must be essentially shorter. The model must be explained in brief with the references to your previous paper, and then the essential results must be presented.

Author Response

Thank you for your time and for the detailed issues with this paper.  
  • Figure 1 is equivalent to Figure 1 in your previous paper [14].
  • Figure 3 appears as a part of Figure 5 in your previous paper [14].
  • Equation (1) is equivalent to Equation (2) in your previous paper [14]. The equation defines already defined model, so the reference is needed.
Indeed the figures and equations are equivalent, they have been removed and replaced with a reference to the previous paper, along with any salient information distilled into a sentence to preserve understanding and meaning.  
  • Figure 2 is equivalent to Figure 2 in your previous paper [14].
  • Figure 4 mimics Figure 4 in your previous paper [14] with less details.
The figures depicting the method and how they integrate into the control loop were deemed too important to remove outright. As such they have been redesigned and simplified to preserve the underling meaning, without reusing the figures from the previous paper.  
  • Figure 7 presents the same trajectories as Figure 7 in your previous paper [14].
  • Figure 8 (right) is equivalent to Figure 8(a) in your previous paper [14].
  • Figure 11 (right) is equivalent to Figure 9(a) in your previous paper [14].
For the experimental results we agree with the reviewer, for the lack of novelty of the results when compared with the previous paper, as they used the same dataset. In order to correct this major issue, novel results and analysis were done that are exclusive to this paper.   With the new changes, simplified figure, and preserving the richness of the results, we hope that the shorter and simpler paper is capable of conveying the same message, while staying relevant with respect to the presented results.